# Antimicrobial Peptides in Infectious Diseases and Beyond—A Narrative Review

**DOI:** 10.3390/life13081651

**Published:** 2023-07-28

**Authors:** Petros Ioannou, Stella Baliou, Diamantis P. Kofteridis

**Affiliations:** 1School of Medicine, University of Crete, 71003 Heraklion, Greece; 2Internal Medicine, University Hospital of Heraklion, 71110 Heraklion, Greece

**Keywords:** antimicrobial peptides, infectious diseases, microbiology, antimicrobial resistance, antibiotics

## Abstract

Despite recent medical research and clinical practice developments, the development of antimicrobial resistance (AMR) significantly limits therapeutics for infectious diseases. Thus, novel treatments for infectious diseases, especially in this era of increasing AMR, are urgently needed. There is ongoing research on non-classical therapies for infectious diseases utilizing alternative antimicrobial mechanisms to fight pathogens, such as bacteriophages or antimicrobial peptides (AMPs). AMPs are evolutionarily conserved molecules naturally produced by several organisms, such as plants, insects, marine organisms, and mammals, aiming to protect the host by fighting pathogenic microorganisms. There is ongoing research regarding developing AMPs for clinical use in infectious diseases. Moreover, AMPs have several other non-medical applications in the food industry, such as preservatives, animal husbandry, plant protection, and aquaculture. This review focuses on AMPs, their origins, biology, structure, mechanisms of action, non-medical applications, and clinical applications in infectious diseases.

## 1. Introduction

Despite significant recent developments in medical research and clinical practice, therapeutics for infectious diseases are greatly limited by the emergence of antimicrobial resistance (AMR) [1,2,3]. AMR is one of the top threats in medicine and directly threatens society with increasing rates of morbidity, mortality, and healthcare costs [4,5,6,7,8,9,10]. For example, more than 2.8 million antimicrobial-resistant infections occur each year in the USA, resulting in more than 35,000 deaths [5]. AMR is also associated with a high cost for healthcare and society. For example, it is estimated that by 2050, the approximate annual worldwide cost of AMR could be between USD 300 billion and 1 trillion [7,11].

From a clinical perspective, infections by antibiotic-resistant bacteria develop in patients with previous antimicrobial use, recent hospitalization, or those who live in long-term care facilities [12,13,14,15,16]. The mechanism of development of AMR involves the exposure of bacteria in the patient’s microbiome to antibiotics, the elimination of bacteria that are sensitive to the used antibiotic, and the selection of those who are resistant [17,18,19]. Then, the resistant bacteria could be transmitted to other people via direct or indirect contact. This could lead to the spread of AMR in the hospital environment and long-term care facilities [12,13,14,15,16]. The development of antimicrobial resistance involves several mechanisms, such as limiting the uptake of antibiotics, the enzymatic inactivation of the drug, modification of the drug target, or the production of efflux pumps that will lead to the removal of the antibiotic [17,20,21]. Importantly, AMR is not only associated with using antimicrobials in humans. For example, an increased concentration of antimicrobials in industrially polluted water, untreated hospital effluent, municipal sewage, and in rivers and seas may also drive the development of AMR in environmental bacteria [22,23,24,25,26]. Beyond the selective pressure of antibiotics on susceptible microorganisms, other mechanisms are associated with the spread of AMR. These involve the mobilization and transfer of genetic elements from resistant microorganisms to susceptible ones [22,27]. The microorganisms spreading the AMR could be of nosocomial or environmental origin, such as those thriving in the external environment, in animals, or humans [22,27].

Among the more-common antimicrobial-resistant microorganisms are carbapenem-resistant *Acinetobacter*, carbapenem-resistant *Enterobacteriaceae* (CRE), methicillin-resistant *Staphylococcus aureus* (MRSA), and vancomycin-resistant *Enterococcus* (VRE) [5,28,29]. Currently, the clinical practice of infectious diseases and clinical microbiology is greatly limited because of AMR. The existing therapeutic approach involves using parenteral antimicrobials as single drugs or in combination, targeting the offending microorganism. Treatment is usually started empirically and modified according to culture and antimicrobial susceptibility results. Guidelines published by international and local infectious disease societies aid in selecting the suitable treatment, even for multidrug and extensively drug-resistant microorganisms [30,31,32].

However, in many cases, especially regarding pan-drug-resistant *A. baumannii*, *Pseudomonas*, or *Klebsiella*, no current available single antimicrobial agent retains in vitro susceptibility. In that case, treatment involves providing antimicrobial combinations that have in vitro synergy in an attempt to reduce morbidity and mortality [33,34,35]. Due to the mediocre activity of this approach, and the lack of viable antimicrobial treatment options, there is an unmet need for developing newer therapeutic options for these difficult-to-treat pathogens. Even though the antimicrobial pipeline contains several tested antimicrobials, very few complete all necessary steps toward approval for clinical practice. Even in that case, however, up until now, most of these antimicrobials belong to already existing antimicrobial classes [36,37]. Even so, unfortunately, resistance quickly develops, thus leaving few or no options for treatment [38,39,40]. To tackle antimicrobial resistance, the discovery of new antibiotics is not enough. Infection control practices, along with antimicrobial stewardship interventions, are also required. Infection control practices could lead to a reduction in the transmission of resistant microorganisms between humans. These measures are primarily in practice in healthcare settings, involving, among other things, appropriate guidance on disinfection and contact, droplet, or airborne isolation, depending on the pathogen [41,42]. On the other hand, antimicrobial stewardship interventions allow for the better use of antimicrobials by reducing the unnecessary exposure of patients to these drugs. The optimization of antimicrobial use through educational approaches, formulary restriction, solicited or even unsolicited infectious disease consultations, and prospective audits and feedback could lead to reduced AMR without compromising patient safety [43,44,45].

Novel treatments for infectious diseases in the era of increasing AMR are urgently needed. To that end, there is ongoing research in non-classical therapies for infectious diseases, utilizing other antimicrobial mechanisms to fight pathogens, such as bacteriophages or antimicrobial peptides [36,46,47,48,49]. This review focuses on antimicrobial peptides (AMPs) and the ongoing research regarding their development. Moreover, it provides a comprehensive overview of clinical studies where AMPs are used in the fight against infectious diseases, as well as other non-medical applications of AMPs.

## 2. Search Strategy

For the conduction of the present review, PubMed and Scopus were searched with the term ‘antimicrobial peptides’ until 20 June 2023 for original articles providing information on biology, clinical trials, and other medical and non-medical applications of AMPs, as well as for informative reviews on the topic. The resulting articles’ references were also searched to identify additional relevant articles. The extracted information included data on the biology of AMPs, their mechanism of action, non-medical applications, and clinical data from trials in humans.

## 3. Definitions and Origins of AMPs

AMPs are evolutionarily conserved molecules produced naturally by several organisms aiming to combat pathogenic microorganisms. They are an essential component of the innate immune system of several species, such as insects, amphibians, marine organisms, and even mammals. At the same time, plants and bacteria can also produce AMPs [50]. Indeed, invertebrates that lack an adaptive immune system mainly depend on AMPs to fight invading pathogens and eliminate infections [51,52]. Lysozyme, probably the first AMP ever described, was discovered about 100 years ago by Alexander Fleming; however, most of the discoveries in the field of AMPs were made after 1990 [53]. Until now, thousands of AMPs have been described, with some of them being natural and some of them being synthetic [54].

AMPs may be effective against various microorganisms, causing the rapid killing of Gram-positive and Gram-negative microorganisms, fungi, parasites, viruses, or even neoplastic cells. They may even be able to inhibit the growth of antimicrobial-resistant microorganisms with mechanisms that are distinct from those of currently used antimicrobials [55,56,57,58,59]. For example, persulcatusin isolated from ticks (*Ixodes persulcatus*) has antimicrobial activity against MRSA and vancomycin-resistant *S. aureus* (VRSA) [60,61]. Another example of AMPs that are clinically in use, and are of increasing importance in the era of AMR, is polymyxins—cationic polypeptides bound to fatty acids, [62]. They act by binding to the negatively charged part of the lipopolysaccharide (LPS), thus leading to permeability changes in the bacterial cell membrane and cell death [63,64]. Furthermore, many AMPs exert immunomodulatory activity, thereby promoting pathogen eradication in an indirect manner [65,66].

## 4. Structure and Function of AMPs

AMPs generally consist of 10 to 50 amino-acid residues (even though they could have as few as 5 and up to more than 100 amino-acid residues), have a positive net charge, and typically contain a significant number of hydrophobic residues (in general, about 50%) as well as hydrophilic segments, thus possessing amphiphilic characteristics [54,55,67,68,69,70]. They are classified according to their source and origin (from prokaryotes, protista, fungi, plants, and animals), their biological activity (against bacteria, viruses, parasites, fungi, insects, and tumors), their characteristics (such as charge and length), or other factors, such as their chemical modifications, their 3D structure, their binding, and their mechanism of action [71]. The classification of AMPs has to do with their secondary structure, according to which they are divided into three categories: a-helical, b-sheet, or extended/random-coil structure, with most AMPs belonging to the first two categories [56,68,70,72,73]. However, it is notable that the structure may depend on the identity of the AMP and the surrounding environment. Moreover, their structure may vary upon interaction with zwitterion and anionic membranes, thus making this classification problematic [74,75].

As stated initially in architecture, the following quote also applies to biology: “form ever follows function” [76]. Indeed, the mechanism of action of AMPs is closely related to their structural characteristics and physical features, which are mainly characterized by a net positive charge and hydrophobicity [70,77,78]. The presence of many positively charged amino-acid residues such as lysine and arginine provides a positive net charge to AMPs, allowing them to interact with the negatively charged outer cell membrane of bacteria while avoiding interaction with the uncharged eukaryotic cell membrane [79]. Hence, efforts to improve the activity of AMPs aim to replace amino-acid residues with positively charged ones [80,81]. An example is reflected in a study where cathelicidin-BF15 was modified by replacing amino-acid residues, leading to the production of ZY13—an AMP with improved antimicrobial activity, increased stability, and reduced hemolytic potential [82]. Mounting evidence suggests that increasing net cationic charge correlates to the increased antimicrobial activity of AMPs. However, this may also be associated with an increase in hemolytic activity [83,84,85,86,87]. On the other hand, the hydrophobicity of AMPs is a critical characteristic, as most of their amino-acid residues are hydrophobic. This allows AMPs to interact with cell membranes, even though it also adds to their hemolytic activity, as AMPs that are highly hydrophobic penetrate erythrocytic membranes to a higher extent [88,89].

The mechanism of action of AMPs has been a focus of research for many years, since a better understanding of the way these molecules act could allow for the optimization of their function by modifying their primary, secondary, and tertiary structure [70,79,90,91]. AMPs can be categorized based on their mechanism of action into those that kill by acting on the membrane causing disruption and those that act by non-membrane-disruptive mechanisms. The membrane targeting of AMPs can be accomplished through receptor association or without receptors. AMPs cross the membrane after achieving a particular threshold concentration and perturb bacterial membrane permeability through pore opening. More specifically, AMPs that act by membrane-disrupting mechanisms do so by forming a toroidal pore or a barrel-stave, or via a carpet-like mechanism, with all these causing membrane disruption that leads to the leak of intracellular contents, thus causing cell death (Figure 1) [70,92]. In all cases, AMPs accumulate and are appropriately organized on the target cell membrane, a process driven by their hydrophobic nature and the electrostatic interaction of their positive net charge with the negatively charged phospholipids on the cell membrane of the target cell [93]. The peptide–lipid ratio exerts a determining effect on the delivery of AMPs on the bacterial membrane. For example, AMPs are transferred in parallel into the bacterial membrane at a low peptide–lipid ratio [94].

In contrast, AMPs penetrate vertically into the bacterial membrane in the case of a high peptide–lipid ratio. In some instances, AMPs bind to receptors, exerting their efficacy in vitro in the nanomolar range [95]. Alternatively, AMPs display their bacterial potential through interactions with membrane components other than receptors in vitro at micromolar concentrations, thereby causing membrane disruption [96]. The toroidal pore mechanism disrupts the target cell membrane by perpendicularly inserting AMPs into the lipid bilayer. The carpet-like mechanism involves the accumulation of AMPs on the target cell membrane, leading to the penetration of the membrane. Finally, the barrel-stave mechanism, which is more typical for the AMPs that have a helical secondary structure, involves the formation of hydrophilic pores on the target cell membrane, which leads to the release of its intracellular content [51,93,97,98]. AMPs that act via non-membrane-disrupting mechanisms target microbial processes that do not involve the membrane and may resemble the action of classic antimicrobials, such as DNA synthesis, protein synthesis, specific enzyme inhibition, cell wall biosynthesis, inhibiting cell division, inhibiting RNA or DNA functions, and triggering apoptosis through reactive oxygen species (ROS) generation (Figure 1) [55,99,100,101,102].

Furthermore, beyond causing direct microorganism killing, AMPs may also have immunomodulatory properties (Figure 2). They are secreted by many different innate immune system cells, such as neutrophils and macrophages. They can intervene in the inflammatory milieu, leading to a more-controlled secretion of proinflammatory cytokines, preventing excessive tissue damage. At the same time, they also promote angiogenesis and reduce the excessive production of ROS [66,103,104,105]. More specifically, it has been suggested that AMPs are significant factors that attract and activate different populations of immune cells (e.g., polymorphonuclear cells or macrophages) through chemokines. Moreover, AMPs may suppress the expression of several pro-inflammatory cytokines, exert anti-endotoxin activity, and enhance the microbial killing mechanisms of immune cells, such as phagocytosis or the production of neutrophil extracellular traps (NETs) [106,107]. In other cases, AMPs exert their action through the alteration of signaling pathways such as p38, extracellular signal-regulated kinase 1/2 (Erk1/2), c-Jun N-terminal kinase JNK mitogen-activated protein (MAP)-kinases, Nuclear factor kappa-light-chain-enhancer of activated B cells (NF-kB), or Phosphoinositide 3-kinase (PI3K)/Protein kinase B (Akt), independently of bacterial ligand binding [108]. On the other hand, some AMPs, such as LL37, can induce auto-inflammation through the recognition of self-DNA by TLR9, self-RNA by TLR7/8, or double-stranded RNA by TLR3 [106]. In this regard, LL-37 has been reported to be harmful, allowing mast cell degranulation and consequent histamine release, which in turn leads to vasodilation [109] through the activation of signaling pathways such as PI3K/Akt, Erk1/2, and JNK [110]. Examples of the mechanism of action of AMPs are shown in Table 1 and in more detail in other studies [111,112].

## 5. The Physiological Role of AMPs

Physiologically, AMPs are either produced continuously or readily synthesized in response to a perceived microbial threat of the host by the upregulation of the corresponding genes. As an example, in humans, psoriasin, lactoferrin, and dermcidin are continuously present in the skin, while cathelicidin LL-37 is readily produced in response to infection [128,129]. In vertebrates, AMPs have the potential to kill microorganisms directly; however, they can also modulate the immune system by activating and recruiting cells of the immune system during infection [70,97,130]. Many different types of AMPs have been described from cells of the immune system, secretions, and epithelia of animals and amphibians [131,132]. More than 30 cathelicidins have been identified and described in mammals. They are stored in inactive forms in granules of the neutrophils so that after neutrophil activation, they can be released in an active state, contributing to microorganism eradication [133,134]. Being produced in other vertebrates, not only in mammals, several AMP classes other than cathelicidins have been described and have been reported to exert their mechanisms of action, such as modulation of the immune response, reduction of inflammation, and tissue damage, as well as the destruction of invading microorganisms [135,136,137,138,139]. In terms of antimicrobial activity, AMPs are very effective against bacteria, viruses, and fungi [139,140].

Insects and plants also produce AMPs, and these are of increased significance for these organisms since they lack an adaptive immune response and they only rely on the response of their innate immune system to combat invading pathogens [70]. Many AMPs have been isolated and described from insects, and more specifically from their hemolymphs, hemocytes, and epithelial cells [141,142,143]. In plants, leaves, seeds, roots, and tubers of plants have been found to produce AMPs, with defensins and thionins being some notable examples [103,144,145]. They display high thermal, proteolytic, and chemical stability due to the multiple disulfide bonds they contain due to the numerous cysteine residues [103,144,145,146]. The mechanism of action of AMPs produced by plants involves the disruption of the cell membrane and pore formation, the targeting of protein synthesis, and also the targeting of the DNA synthesis of pathogens [147]. Some of the AMPs produced by insects are defensins, cecropins, diptericins, and drosocins, and they exert similar activity against Gram-positive and Gram-negative microorganisms and fungi. One example of their antibacterial mechanism of action is associated with their cationic nature, which leads to the permeabilization of the bacterial cell membranes [148,149,150,151].

AMPs are also crucial for bacteria since their production allows them to adapt and survive in antagonistic environments where competition with other microorganisms in specific environmental conditions threatens their existence [97]. Many AMPs have been isolated from bacteria and present significant antimicrobial activity against Gram-positive, Gram-negative microorganisms and fungi. These AMPs, also called bacteriocins, are physiologically produced by those bacteria to inhibit or kill microorganisms that antagonize them without causing toxicity to the bacterium that produces them, thus providing a survival benefit [70,152].

## 6. Applications of AMPs

### 6.1. Food Industry

One of the most promising applications of AMPs is their potential as alternative regimens to antimicrobial treatment. This will be described in detail later on. Other non-medical uses include their use in the food industry as preservatives [153,154]. Such examples include nisin, an AMP consisting of 34 amino-acid residues, which is isolated from *Lactococcus lactis* and used as a preservative in skim and whole-fat milk, cottage cheese, milk putting and other dairy products; enterocin, which is isolated from *Enterococcus faecium*, aureocin, an AMP isolated from *S. aureus*; bovicin, which is isolated from *Streptococcus bovis*; reuterin, an AMP isolated from *Lactobacillus reuteri*; and others that are mainly used as preservatives in dairy products [154]. Lactoferrin is another example of an AMP, commonly used as an antimicrobial agent in the USA to preserve meat products. At the same time, its derivative, lactoferricin, is an AMP with more potent antimicrobial activity and relative heat resistance, which provides higher food preservation capacity [155,156]. Another example of an AMP used as a food preservative is natamycin, commonly applied on the surface of cheese and salami-type sausages. Natamycin is produced by *Streptomyces* spp. and has potent antimicrobial activity against the majority of food-borne fungi even though it has no activity against bacteria or viruses [157]. Spheniscin is a defensin produced in the stomach of king penguins and preserves undigested food for future use in the egg incubation period. Thus, it was hypothesized that it could be of use as a food preservative [158].

### 6.2. Animal Husbandry

AMPs are also crucial in animal husbandry. Due to their potential for a strong antimicrobial effect with a low likelihood of resistance, it was considered that they could maintain sustainable livestock production. For example, the dietary supplementation of chickens with nisin changed the intestinal microbiome of chickens, leading to fewer counts of *Enterobacterales* and *Bacteroides* [159]. One step further, transgenic animals, exploiting genetic manipulation techniques, can produce AMPs that can optimize their production and reduce the possibility of infection. For example, the expression of bovine or human lactoferricin in the mammary glands of transgenic goats provided a wide range of antimicrobial activities [160]. Dietary supplementation with AMPs to reduce outbreaks of animal disease and improve growth performance and animal health in animal husbandry was proposed as an alternative to dietary supplementation with antimicrobials until it was banned by the European Union [156,161]. The addition of AMPs such as colistin E1, cipB-lactoferricin-lactoferrampin, and cecropin AD to the diets of weanling pigs led to an improvement of growth and health condition by beneficially altering the gut microbiome [162,163,164]. Another vital use of AMPs in the context of animal husbandry has to do with their antiviral activities, such as against porcine epidemic diarrhea virus, infectious bronchitis virus (IBV), porcine transmissible gastroenteritis virus, and influenza A [165,166,167,168]. Hence, AMPs could be used in animals to reduce the dangers associated with a viral infection, as exemplified in the case of the infection of chick embryos treated with swine intestine AMP, which has potent antiviral activity against IBV [166].

### 6.3. Plant Protection

Plant protection is another non-medical application of AMPs. AMPs could replace pesticides in the fight against plant pathogens and insects, which could allow for the reduction of environmental pollution and the minimization of risk for human health harms associated with the use of pesticides [169]. AMPs such as iseganan or pexiganan display important activity against phytopathogenic bacteria such as *Pectobacterium* spp., while other AMPs have important antifungal activities that could be used for plant protection [156,170]. The genetic manipulation of plants leading to the recombinant expression of AMPs in their bodies could lead to resistance to phytopathogens. For example, the gene expressing Alf-AFP defensin was introduced in the potato, leading to greater protection against *Verticillium dahliae* [171]. Furthermore, the expression of the mammalian AMP cecropin P1 in transgenic tobacco led to increased plant resistance towards *E. carotovora*, *Pseudomonas marginata*, and *Pseudomonas syringae* [172]. Finally, when an analog of magainin, MSI-99, was expressed in different plants, it provided protection against many bacterial and fungal phytopathogens [173,174,175].

### 6.4. Aquaculture

Finally, AMPs have also been used in aquaculture. Since fish and other sea products are essential components of the human diet, aquaculture is a growing finance sector. However, outbreaks of microbial disease can be associated with a significant delay in production and critical financial losses [156]. To that end, the synthetic AMP epinecidin-1 could hinder the growth of multiple bacteria that are known to be harmful to aquacultural organisms, such as *Pasteurella multocida*, *E. coli*, *Aeromonas* spp., and *Vibrio* spp. [176]. In another case, several AMPs like caerin 1.1, or sole NKLP27, had potent antimicrobial activity against most human and fish pathogenic bacteria. At the same time, they were also able to inactivate many fish viruses that are important in aquaculture [177].

## 7. Potential for the Utilization of AMPs in Infectious Diseases

AMPs could be used alone or in combination with conventional antimicrobials against microorganisms, including difficult-to-treat (DTT) microorganisms, such as multi-drug-resistant (MDR) or extensively drug-resistant (XDR) bacteria. They have the potential for rapidly killing microorganisms with a low possibility of the further development of resistance, which makes them ideal candidates for the fight against resistant pathogens [178,179,180]. An essential difference between AMPs and classical antimicrobials is that the latter target one single biological process. In contrast, AMPs usually exert multiple effects that may involve the disruption of the microbial cell membrane, transcription, translation, protein synthesis, and DNA synthesis [70,181,182,183]. Based on naturally occurring AMPs, synthetic AMPs with optimized pharmacological properties have been designed. These molecules already have increased potency against DTT microorganisms—an approach more effective than the current pharmacological approach of producing classic antimicrobial agents with broad-spectrum activity that is susceptible to the rapid development of AMR [70,182].

## 8. Examples of AMPs with Potential Interest in Infectious Diseases

Many AMPs have been extensively studied to identify whether they could be used in clinical practice in infectious diseases, either in their natural form or after modifications. AMPs belonging to the cathelicidin group are found in animals and humans. They could, theoretically, be suitable for clinical use due to their broad-spectrum antimicrobial activity against bacteria, viruses, and fungi and their immunomodulatory potential [133,184]. For example, cathelicidin LL-37 is one of the most-studied AMPs, exhibiting potent antimicrobial activity that spans bacteria and fungi [185]. LL-37 has been used as a basis for the design of synthetic AMPs with improved antimicrobial activity against DTT pathogens and biofilms [186,187]. For example, SAAP-148, an AMP subjected to modifications of the C-terminal chain of LL-37, has been reported to exert significant microbicidal activity against many DTT pathogens such as *A. baumannii*, *P. aeruginosa*, members of *Enterobacterales* and Gram-positive microorganisms such as *E. faecium* and *S. aureus* [188]. Notably, this AMP was also shown to be active against biofilms of some of these pathogens, such as *A. baumannii*, *P. aeruginosa*, and *S. aureus*, and was able to kill without the selection of resistance [188].

AMPs approved for clinical use nowadays due to their antimicrobial action include cyclic peptides such as polymyxins and gramicidins. Polymyxins were mainly used locally to treat ocular infections and also for the selective decontamination of the gastrointestinal tract. However, the increased AMR of Gram-negative microorganisms has led to a revival of the systematic use of this drug for DTT pathogens such as *A. baumannii*, *P. aeruginosa*, and *Enterobacterales* [64,189]. Gramicidins cause hemolysis to a significant extent, and due to this limitation, they are not used systemically but are primarily administered locally for the treatment of superficial wound infections, as well as throat, nasal and ocular infections [190]. Daptomycin, on the other hand, is a cyclic AMP already approved for systematic use, for the treatment of skin and soft tissue infections by Gram-positive bacteria [191].

Several other AMPs targeting multiple microorganisms, such as pexiganan and omiganan, are under evaluation in clinical trials. Pexiganan has been studied as a topical cream in phase III clinical trials for treating infection in diabetic patients with ulcers. Moreover, there are planned studies for the evaluation of its use in complicated skin and soft tissue infections [56,191,192]. Omiganan, an analog of indolicidin with activity against Gram-positive and Gram-negative bacteria and fungi, has been examined in clinical trials as a local treatment for infections associated with catheters, genital warts, and other conditions [56,191,193].

Other AMPs have also been studied in clinical studies of other conditions, related to their antimicrobial effect. For example, in a study in patients with periodontitis, AMP Nal-P-143 was found to be able to inhibit periodontal pathogens such as *Fusobacterium nucleatum, Streptococcus gordonii*, *Treponema denticola*, and *Porphyromonas gingivalis* and improve periodontal status [194].

## 9. Comparison of AMPs with Classic Antibiotics

There are several advantages of AMPs compared to classic antibiotics that encourage using them as therapeutic options. First of all, the primary clinical challenge of traditional antibiotics is the development of resistance. The potential of bacteria to convey resistance to AMPs is shallow, given that they are usually released during immune responses to combat microbial infections [195]. For example, a systematic study has reported that resistance against AMPs after point mutations and gene amplification was negligible. In contrast, cross-resistance between AMPs and antibiotic-resistant bacteria has not been observed [196]. Moreover, the mechanisms of the killing of AMPs are unique and may bypass AMR mechanisms. At the same time, their potential to activate the immune system provides an extra function not elaborated by classic antibiotics.

Another important antibacterial mechanism of AMPs is their biofilm inhibitory property [197]. AMPs hold great promise as a new-generation antimicrobial treatment. For these reasons, the Food and Drug Administration (FDA) has proceeded to approve three AMPs, while three more are being examined for potential clinical application [198]. Along with the above, several reports have highlighted that AMPs are less susceptible to developing host concentration-dependent toxicity. Significantly, AMPs can attenuate sepsis by neutralizing endotoxins, as shown in both in vitro and in vivo settings [199,200]. In contrast, antibiotics can potentiate septic shock due to the increased secretion of pathogen-associated molecular patterns (PAMPs), thereby aggravating infection severity [201,202]. Last but not least, the good thermal stability and water solubility of AMPs, their reduced bacterial mutagenesis, their subjection to bioengineering techniques, and their potential use in immunocompromised individuals constitute other advantages that support their clinical application compared to conventional antibiotics [54].

Despite the reported advantages of AMPs, several shortcomings of AMPs hinder their clinical application, such as high extraction costs [203], poor bioavailability [204], short half-lives [205], cytotoxicity, and lack of specificity [206]. Regarding AMPs’ toxicity, it is notable that toxicity can be either cellular or systemic [207]. The cellular cytotoxicity originates from their adopted amphiphilic structure because AMPs interact in a non-specific manner [51]. The systemic toxicity of AMPs can be attributed to the following: either sustained immune response or the uncontrolled function of the central nervous system due to their penetration of the blood–brain barrier or obstruction of blood vessels, leading to blood coagulation [207]. As a result, the topical administration of AMPs may be the safest mode of delivery to increase the benefit-to-harm ratio. Another last major limiting factor is their possible instability due to their degradation by proteases [208,209].

To address the challenges above, several strategies have been developed to improve efficacy. Chemical modifications have been proposed to circumvent the potential protease cleavage of AMPs. Specifically, the use of D-amino acids, cyclization, acetylation, dimerization, methylation, and peptidomimetics are some suggested chemical modifications of AMPs that sustain their antimicrobial potential while maintaining their stability [210,211,212,213]. On the other hand, the incorporation of AMPs into drug nano-delivery vehicles contributes to the excellent bioavailability of AMPs, as indicated by their low kidney clearance and their consequent increased retention [214], enabling their controlled release to target sites. Apart from this, nano-delivery systems offer improved pharmacokinetic properties, safety, and stability to AMPs, thereby increasing their efficacy [215,216]. In particular, several nanostructures have been deployed for the delivery of AMPs, including liposomes, micelles, dendrimers, liquid crystalline systems, hydrogels, polymeric nanoparticles, microspheres, metal nanocrystalline materials, carbon nanotubes, quantum dots, mesoporous silica nanoparticles and nano-fibers [217].

## 10. Combination of AMPs with Classic Antibiotics or Other Molecules

Due to the challenges encountered using a single antimicrobial, especially in pathogens with significant AMR [218,219], the combination of AMPs with either AMPs or antibiotics emerges as a promising solution (Table 2 and Table 3) [220]. The rationale for combining AMPs involves the additivity or synergy of their molecular mechanisms due to the opening of bacterial pores for extended periods, the blocking of pore repair, or disturbing bacterial intracellular functions to a greater extent. Similarly, AMPs could increase the therapeutic efficacy of antibiotics through the better delivery of antibiotics into bacterial cells, granting them access to their intracellular targets. The effectiveness of antimicrobial synergies may also involve targeting multiple independent signaling pathways in microorganisms. For example, combinations of antimicrobials present strong antimicrobial efficacy, circumventing the potential of bacterial drug resistance and possible concentration-associated toxicity [221]. Last, such synergistic activities of AMPs can also arise with other compounds or histones (Table 4 and Table 5). Several examples of AMP combinations that display a positive synergistic effect are described in Table 2, Table 3, Table 4 and Table 5.

## 11. Examples of AMP Use in Clinical Practice

Although AMPs are abundant, few have been selected through preclinical studies to proceed to clinical trials for potential use in humans. One such example is daptomycin. Daptomycin is a cyclic 13-member lipopeptide that is produced by *Streptomyces roseosporus* and was discovered in the 1980s [260]. The drug was not approved for clinical use initially since it was used twice daily, leading to muscle toxicity, despite its clinical efficacy. After changing the dose to once daily, the drug was successfully granted application, as the adverse events were drastically reduced [260]. Its mechanism of action is unclear compared to other antibiotics, but it resembles that of a classic AMP. It targets the bacterial cell membrane in a calcium-dependent manner, preferably at the division septum, and disrupts the architecture of the membrane after the oligomerization of the drug [261,262,263]. Randomized clinical trials have evaluated the activity of daptomycin in acute bacterial skin and skin structure infections (ABSSSIs) in a once-daily dose compared to the standard of care (vancomycin and antistaphylococcal penicillins) and showed they were comparable [264,265]. Thus, the FDA approved daptomycin for clinical use for the treatment of Gram-positive cocci, such as *S. aureus* (including strains resistant to methicillin), *Enterococcus faecalis*, and streptococci. In contrast, currently, this drug is used for many other infections caused by Gram-positive microorganisms [266,267,268].

Colistin is a cationic AMP and belongs to the family of polymyxins [269]. These compounds are among the oldest antibiotics, since they were discovered in 1947, and they were used for infections by Gram-negative microorganisms until other potent antimicrobials with fewer toxicities were discovered and licensed for clinical use, such as aminoglycosides [260]. Their mechanism of action involves the penetration of the cell membranes of Gram-negative bacteria, electrostatic interaction with phospholipids, and the disruption of membranes. Moreover, they can bind to the cell wall of lipopolysaccharide (LPS) and block some of its biological functions [270]. In recent decades, clinicians started using colistin again due to the emergence of XDR pathogens, such as *P. aeruginosa* or *A. baumannii* [33,269,271]. To that end, colistin has been successfully used in patients with highly resistant pathogens in many different infections and has shown adequate activity either as monotherapy or, more commonly, in the context of treatment with combinations of antimicrobials [272,273,274].

It is of note, though, that few AMPs are currently licensed for clinical use in humans. The majority of AMPs evaluated either in vitro or in vivo may not proceed to clinical studies, or if they do, a minority of them may be granted authorization for clinical use. Table 6 shows a limited number of examples of AMPs that have proceeded to clinical trials in humans, along with the main study results.

## 12. Conclusions

The increasing prevalence of AMR, with the high morbidity and mortality caused by antibiotic-resistant microorganisms, has led to the need to intensify the production of new antibiotics and identify new weapons in the fight against infectious diseases. To that end, AMPs emerge as valuable tools in infectious diseases, even though they have multiple other applications beyond medicine. Several AMPs are already in use in infectious diseases, such as daptomycin, colistin, and bacitracin; several others are under evaluation for potential use in the future. AMPs may have a vital role alone or in combination with other therapeutic options, such as classic antibiotics already in use. The ability of AMPs to directly kill bacteria without being affected by AMR adds an essential mechanism of action for clinicians caring for patients with infections. The results of studies involving AMPs with promising in vitro activity should be confirmed in well-designed, blind, randomized, controlled clinical trials that will evaluate their safety and efficacy in real-world situations, thus allowing their approval by regulatory authorities.

## Figures and Tables

**Figure 1 life-13-01651-f001:**
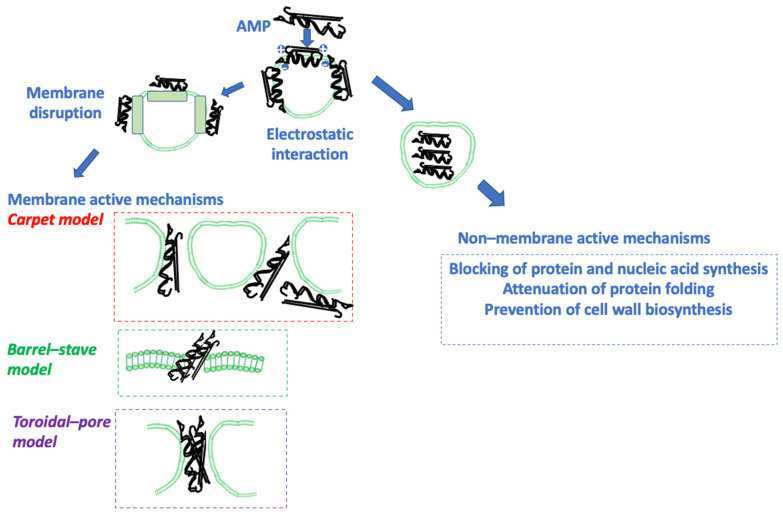
The mechanisms of action of AMPs. AMPs may act through the permeabilization of bacterial cytoplasmic membranes in three different ways. In the carpet model, most AMPs are located on the membrane, causing the perturbation of its integrity and the formation of micelles. In the barrel-stave model, the AMPs are inserted into the membrane, causing the formation of a pore with the hydrophilic side facing the interior of the pore and the hydrophobic side towards the lipid core of the membrane. A transmembrane pore model involves the formation of a pore, as in the toroidal-pore model, but the pore is also lined with the hydrophilic side of the AMPs and the head parts of the membrane phospholipids. AMP: antimicrobial peptide.

**Figure 2 life-13-01651-f002:**
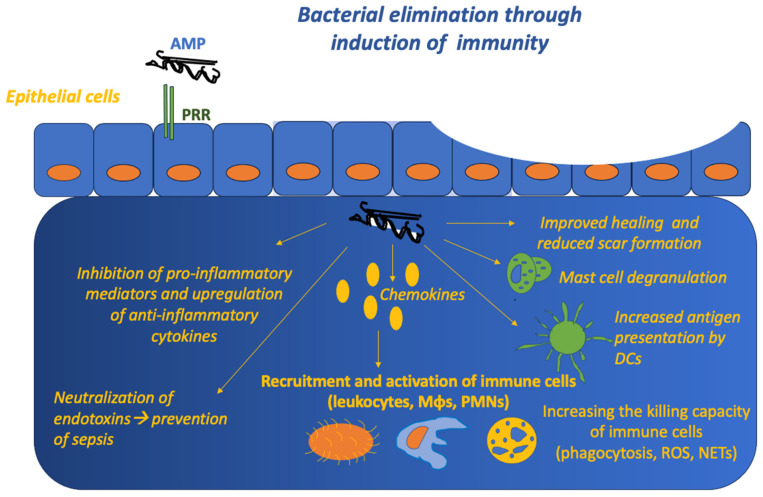
Graphical representation showing the immunomodulatory function of AMPs. Apart from the direct killing mechanisms of AMPs, they can kill pathogens through the induction of innate and adaptive immunity. AMPs can induce cells involved in innate immunity, such as macrophages (Mφs), polymorphonuclear cells (PMNs), and mast cells. AMP: antimicrobial peptide; DCs: dendritic cells; Mφs: macrophages; PRR: pathogen recognition receptor; PMN: polymorphonuclear cell; ROS: reactive oxygen species; NETs: neutrophil extracellular traps.

**Table 1 life-13-01651-t001:** Classification of antimicrobial peptides based on the mechanism of action.

Name	Source	Mechanism of Action	Ref
Abaecin	Bumblebee	Inhibits DnaK	[113]
Alamethicin	Fungus *Trichoderma viride*	Barrel-stave transmembrane pore model	[114]
Apidaecin Hb1a	Honeybee	Inhibits protein biosynthesis by targeting ribosomes, inhibits DnaK and GroEL, inhibits ABC transport system, binds LPS	[115]
Buforin II	Enzymatic cleavage of buforin I (from Asian toad Bufo bufo gargarizans)	Inhibits DNA, inhibits RNA	[100]
Cecropin P1	Pig intestine	Carpet model/Detergent-like mode non-membrane pore model	[116]
Diptericin	Hemolymph of injured Sarcophaga peregrine larva	Inhibits septation	[117]
HD5 (oxidized)	Human small intestine	Inhibits cell division	[118]
Histatin-5	Human saliva	Inhibits proteases, inhibits MMP-2 and MMP-9, inhibits generation of ROS	[119]
Lacticin Q	*Lactococcus lactis* QU 5	Toroidal transmembrane pore model	[120]
Magainin 1	African clawed frog Xenopus laevis	Inhibits energy metabolism proteins, inhibits amino-acid metabolism	[121]
MBI-27	Derived from part of silk moth cecropin and bee melittin peptides	Inhibits LPS	[122]
Nisin	Lactococcus lactis	Inhibits lipid II in peptidoglycan biosynthesis	[123]
Ostricacin-1	Ostrich defensin	Inhibits DNA	[124]
Tachyplesin	Horseshoe crab hemocytes	Binds DNA minor groove	[125]
Thanatin	*Podisus maculiventris*	Agglutination non-membrane pore model	[126]
tPMP-1	Platelet granules	Activation of autolytic enzyme	[127]

AMPs: antimicrobial peptides; LPS: lipopolysaccharide; ROS: reactive oxygen species. This table is not exhaustive of all AMPs or all mechanisms of action.

**Table 2 life-13-01651-t002:** Examples of AMP combinations with antibiotics that exert positive synergistic effects.

AMP	Antibiotic	Target	Ref
Tridecaptin M	Rifampicin, vancomycin, ceftazidime	*A. baumannii*	[222]
Lactoferricin	Ciprofloxacin, ceftazidime	*P. aeruginosa*	[223]
LL-37, HBD3	Tigecycline, moxifloxacin, piperacillin/tazobactam, meropenem	*C. difficile*	[224]
P10	Ceftazidime, doripenem	*A. baumannii* and *P. aeruginosa*	[225]
Gad-1	Kanamycin, ciprofloxacin	*P. aeruginosa*	[226]
Nisin	Penicillin, chloramphenicol, ciprofloxacin, indolicidin, or azithromycin	*S. aureus*	[227,228]
SAAP-148	Demeclocycline hydrochloride (DMCT)	*P. aeruginosa*	[229]
(SLAP)-S25	Cefepime, colistin, ofloxacin, rifampicin, tetracycline, and vancomycin	multidrug-resistant Gram-negative pathogens	[230]
Colistin	tigecycline, carbapenem, gentamicin	*Klebsiella* KPC	[231,232]
Octaarginine	Vancomycin	biofilms of *S. aureus*	[233,234]
Sphistin, Sph12−38	Rifampicin, azithromycin	*P. aeruginosa*	[235]
DP7	Azithromycin, vancomycin	*S. aureus*, *P. aeruginosa*, *A. baumannii*, *E. coli*	[236]
P10	Ceftazidime, doripenem	*A. baumannii*, colistin-resistant *P. aeruginosa*	[225]
Melittin	Doripenem and ceftazidime	*A. baumannii* and *P. aeruginosa*	[237]
LL 17–29	Chloramphenicol	*S. aureus*, *P. aeruginosa*	[218]
Nisin Z, pediocin, or colistin	Penicillin, ampicillin, or rifampicin	*P. fluorescens*	[238]
Melamine	Ciprofloxacin, fluoroquinolone	*P. aeruginosa*	[239]
Indolicidin, polymyxin B	Tobramycin, gentamycin, and amikacin	*P. aeruginosa*	[240]
Arenicin-1	Ampicillin, erythromycin, and chloramphenicol	*S. aureus*, *S. epidermis*, *P. aeruginosa*, and *E. coli*	[241]

AMP: antimicrobial peptide.

**Table 3 life-13-01651-t003:** Examples of AMP combinations that exert positive synergistic effects.

AMP	Synergistic AMP Molecule	Target	Ref
HsAFP1	RsAFP2 and RsAFP1	Biofilm cells	[242]
Diptericins	Attacins	*P. burhodogranariea*	[243]
Pexiganan	Melittin	*S. aureus*	[244]
Abaecin	Hymenoptaecin	Gram-negative bacteria	[245]
Esculentin-1a	Aztreonam	*P. aeruginosa*	[246]
PGLa	Magainin 2	*E. coli*	[247,248,249]
VG16KRKP	KYE28	plant pathogens	[250]
Nisin A	Epsilon-poly-L-lysine	*Bacillus cereus* and *L. monocytogenes*	[251]
Nisin	Colistin	Biofilms of *P. aeruginosa*	[225,252]
Nisin	Colistin	*A. baumannii*, colistin-resistant *P. aeruginosa*	[225]
Magainin-2	Peptidyl-glycylleucine-carboxyamide	*E. coli*	[253]
Indolicidin, LL-37	Bactenecin	*P. aeruginosa* and *E. coli*	[254]
Indolicidin	Bactenecin	*P. aeruginosa* and *E. coli*	[254]
Protegrin 1	LL-37	*E. faecalis*	[254]
Bactenecin	LL-37	*E. faecalis*	[254]
Protegrin1	Bactenecin	*E. faecalis*	[254]
Apidaecin, pexiganan	LL 19–27	*E. coli*	[219]
Galleria mellonella anionic peptide 2	Lysozyme	Gram-negative bacteria	[255]
B-defensin, LL-37	Lysozyme	*S. aureus*, *E. coli*	[256]

AMP: antimicrobial peptide.

**Table 4 life-13-01651-t004:** Examples of AMP combinations with other compounds that exert positive synergistic effects.

AMP	Synergistic Compound	Target	Ref
Nisin	Citric acid	*S. aureus* and *L. monocytogenes*	[257]
Nisin A	Epsilon-poly-L-lysine	Gram-positive food-borne pathogens *Bacillus cereus* and *L. monocytogenes*	[251]
Polymyxin B, Gramicidin S	Silver nitrate, silver nanoparticles	Gram-negative bacteria	[258]

AMP: antimicrobial peptide.

**Table 5 life-13-01651-t005:** Examples of AMP combinations with histones that exert positive synergistic effects.

AMP	Histone	Target	Ref
LL-37 and magainin-2	H2A and H3	Gram-positive and Gram-negative bacteria	[259]
Polymyxin B	H2A	*E. coli*	[259]

AMP: antimicrobial peptide.

**Table 6 life-13-01651-t006:** Examples of antimicrobial peptides in human clinical trials.

Study	Population	Intervention	Comparator	Outcome
Fowler et al., 2006 [275]—Phase III RCT	Patients with *S. aureus* bacteremia with or without endocarditis	IV treatment with the AMP daptomycin	IV treatment with low-dose gentamicin plus either an antistaphylococcal penicillin or vancomycin	Daptomycin was non-inferior to standard-of-care. Treatment success rates were similar in subgroups of patients with complicated bacteremia, right-sided endocarditis, and methicillin-resistant S. aureus
Miller et al., 1948 [276]—non-randomized study	130 patients with superficial infections of the skin and 35 patients with secondary skin infections	Bacitracin applied locally	None	Cure rate higher than 50% in superficial skin infections and 100% in secondary skin infections
NCT05340790—Phase I RCT	Healthy female volunteers	Dose 1 to 5 of AMP PL-18 vaginal suppositories	Placebo doses 1 to 5 of vaginal suppositories	Safety assessment (recruiting)
Gronberg et al., 2014 [277]—RCT	Adult patients with hard-to-heal venous leg ulcers	Repeated doses of LL-37 applied locally	Repeated doses of placebo applied locally	Safe and well tolerated. Significant early healing of ulcers
Daley et al., 2017 [278]—Phase III RCT	Adult patients with *Clostridioides difficile* infection	AMP surotomycin orally	Vancomycin orally	Non-inferior but non-superior to vancomycin for clinical response
Lipsky et al., 2008 [192]—Phase III RCT	Adult diabetic patients with infected wounds at the lower extremities	AMP pexiganan locally and oral placebo	Local placebo and oral ofloxacin	Pexiganan is comparable to oral ofloxacin for mildly infected diabetic ulcers
Niemeyer-van der Kolk et al., 2020 [279]—Phase II RCT	Patients 18–65 years old with mild-to-moderate atopic dermatitis	AMP Omiganan locally	Vehicle gel locally (placebo)	Treatment with the AMP improved dysbiosis in mild to moderate atopic dermatitis patients Small improvements in clinical scores were detected
Peek et al., 2020 [280]—Phase II RCT	Adults with chronic suppurative otitis media resistant to antibiotic therapy	AMP P60.4Ac locally with ear drops	Vehicle locally with ear drops (placebo)	Safe and well-tolerated treatment. Significantly higher treatment success than placebo
NCT00231153, Phase III RCT	Patients with central venous catheters	AMP Omiganan 1% gel local application at the catheter insertion site	Povidone-Iodine 10% local application at the catheter insertion site	Failed to show adequate efficacy in catheter-associated infections
NCT04767321, Phase I/II RCT	Adults 18–65 years old with persistent carriage of *S. aureus*	Nasal application of the AMP LTX-109	Nasal application of placebo	Safety, tolerability, and microbial eradication—recruitment completed
Mercer et al., 2020 [281]—Phase I and Phase II RCTs	12, 48, and 47 patients with onychomycosis of the toenail	Local application of NP213	Local application of placebo	NP213 clinical safety profile. Positive patient-reported outcomes
Mullane et al., 2015 [282]—Phase II RCT	72 patients with *Clostridioides difficile* infection	LFF571 orally	Vancomycin orally	The rate of clinical cure was non-inferior to that of vancomycin. Similar 30-day sustained cure rates. More adverse events for LFF571.
Corey et al., 2014 [283]—Phase IIa RCT	84 adult patients with ABSSSI	GSK1322322 orally	Linezolid orally	Clinical success in the ITT population and the per-protocol population were 67 and 91% in the GSK1322322-treated group and 89 and 100% in the linezolid-treated group

ABSSSI: acute bacterial skin and skin structure infection; AMP: antimicrobial peptide; ITT: intention-to-treat; IV: intravenous; RCT: randomized controlled trial; This table may not be exhaustive of all antimicrobial peptides that are currently approved or in clinical trials.

## Data Availability

The data presented in this study are available on request from the corresponding author.

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
