# Peer review of "Antimicrobial Peptides in Infectious Diseases and Beyond—A Narrative Review"

_life, 2023, doi:10.3390/life13081651_

Round 1
Reviewer 1 Report (Previous Reviewer 1)
No doubt, the authors made an exceptional effort to make the paper worthy.
The English language needs to check carefully in the revision stage because of many careless mistakes in many positions.
Authors tried to grab every information in few paragraphs and with general information no such blunt literature found in the introduction section. This manuscript can be accepted after addition of few more references in the introduction section.
Moreover, please check the format of tables.
The English language needs to check carefully in the revision stage because of many careless mistakes in many positions
Author Response
No doubt, the authors made an exceptional effort to make the paper worthy.
Response: Thanks for the comment. The truth is that we made extensive revisions according to the reviewers’ comments. We thank the reviewers for providing the opportunity to improve this manuscript.
The English language needs to check carefully in the revision stage because of many careless mistakes in many positions.
Response: Thanks. The manuscript was revised by a native English speaker and was improved in terms of clarity and correctness. Several corrections can be seen throughout the text in the revised version of the manuscript.
Authors tried to grab every information in few paragraphs and with general information no such blunt literature found in the introduction section. This manuscript can be accepted after addition of few more references in the introduction section.
Response: Thanks. As suggested by the reviewer, we added more references in the introduction section. The revised manuscript now has 49 studies cited in the introduction section. These are 12 more than the 37 studies that were cited in the previous one. The cited literature now better supports the meanings of the introduction section and allows the reader to find more relevant literature for further reading. These changes can be seen in the introduction section of the revised version of the manuscript.
Moreover, please check the format of tables.
Response: The tables were reformatted. We have followed the journal’s instructions for the authors. This can be seen in the revised version of the manuscript.
Reviewer 2 Report (Previous Reviewer 2)
Please include references for each and every example listed in Table 1. Moreover, the authors have completed the majority of the modifications I recommended. Now, the manuscript is eligible for publication consideration.
Author Response
Please include references for each and every example listed in Table 1. Moreover, the authors have completed the majority of the modifications I recommended. Now, the manuscript is eligible for publication consideration.
Response: Thanks for the comment. We have added references for each antimicrobial peptide mentioned in Table 1. This can be seen in the revised version of the manuscript.
Reviewer 3 Report (Previous Reviewer 3)
The modifications are acceptable.
The English language is fine after revision.
Author Response
The modifications are acceptable.
The English language is fine after revision.
Response: We thank the reviewer for the nice comments.
This manuscript is a resubmission of an earlier submission. The following is a list of the peer review reports and author responses from that submission.
Round 1
Reviewer 1 Report
English language of whole paper is poor and difficult to understand, please improve it and then proceed it for peer review. Thank you
Reviewer 2 Report
Comments to the authors:
1. The authors of this study did a great job of discussing the significance of antimicrobial peptides as well as their many benefits. While there is a benefit or an advantage, there is also likely to be a drawback or negative consequence. I would like to advise to the authors that they provide any success stories or instances of the inadequacies of AMPs after following the strategy for improving their efficacy.
2. I would like to suggest to the authors that they address in more detail the factors that contribute to antimicrobial resistance and the methods that may be used to combat it.
3. I would like to advise to the authors that they add success stories or some research examples of instances in which AMPs played a vital role in the control of the disease when all conventional antibiotics failed to do so.
4. Is it possible for the authors to categorize the AMPs (along with examples) depending on their mechanism in the form of a table (Name of the AMP, Source, Usage, and Mechanism of Action)?
5. Tables 3 and 4 are away from the view, make the necessary adjustments.
6. Provide a table listing all AMPs that have been clinically approved and the application.
Reviewer 3 Report
According to the title, Ioannou et al aimed to write a narrative review regarding the use of antimicrobial peptides mainly involved in infectious diseases. Therefore, it was expected that the classification would be designed based on various categories of infectious disorders. Unfortunately, the manuscript is not designed properly and is written in a careless style. It
lacks a special focus and a logical in-depth discussion in each section. This descriptive form with too long (mostly useless description due to poor English) does not increase our knowledge in the field of AMPs and the future of antibiotic therapy. No special conclusion could be derived from this manuscript.
Other comments
Comment 1: There are plenty of reviews about antimicrobial peptides, their origin, structure, and application. What is the novelty of this narrative review?
Comment 2 In line with the previous comment, authors should define the exact keywords, databases, and range of years to retrieve related articles
Comment 3: The English of the manuscript is too poor and not understandable regarding Grammar, clarity, and engagement. The style requires rigorous English editing. Unfortunately, the manuscript is not acceptable in the current format.
What do you mean by “ animals of humans” on page 2, paragraph 2? Or “Even in that case, however, …..”. The term “difficult infections” does not sound scientific.
Most sentences in the manuscript are too long and should be fragmented. This is only one example: “Unfortunately, even though the antimicrobial pipeline contains several antimicrobials that are tested, very few of them successfully complete all necessary steps towards approval for clinical practice, and even those, up until now, mostly belong to already existing antimicrobial classes”
This sophistication is observed especially in section 5
Comment 4: Figure 1 is too preliminary and is not acceptable in the current format. There should be clear and concise details in this level of science for AMP-membrane interaction and immune response requirements. The text which is annotated as figure 2 is not considered a figure!!
Comment 5: Why do you only mention fish throughout the manuscript? Various marine organisms contain AMPs not only fish
Comment 6: Classification of AMPs according to their structure in section 3 is too preliminary and differs depending on the identity of the peptide and the surrounding environment. The structure of peptides might vary upon interaction with zwitterion and anionic membranes which is not considered here.
Comment 7: There are various redundant sentences or concepts in the manuscript (repeated more than once) such as
“As an example, in humans, psoriasin, lactoferrin, and dermcidin are continuously 202 present in the skin, while, cathelicidin LL-37 is readily produced in response to infection”
Comment 8: The conclusion section is not related to the manuscript. It is too general without any take-home message.